



# Using global reanalysis and rainfall-runoff model to study multi-decadal variability in catchment hydrology at the European scale

Pierre Brigode[1], Ludovic Oudin[2]

[1]Univ Rennes, CNRS, Géosciences Rennes - UMR 6118, Rennes F-35000, France

[2]Sorbonne Université, CNRS, EPHE, UMR 7619 METIS, Case 105, 4 place Jussieu, F-75005 Paris, France

*Correspondence to*: Pierre Brigode (pierre.brigode@ens-rennes.fr)

**Abstract.** This study explores the ability of global reanalyses to simulate catchment hydrology at the European scale using a conceptual rainfall–runoff model. We used two reanalyses, NOAA 20CR and ERA-20C, to simulate daily streamflows for over 2000 catchments since the 1840s. Our findings show that both reanalyses perform well, particularly for mean flows,

with simulation performance improving as catchment size increases, though challenges remain for Mediterranean and snow-dominated regions. Additionally, the study highlights significant multi-decadal variations in streamflow, revealing alternating wet and dry periods across Europe. These findings provide valuable insights into long-term hydrological trends and offer a useful framework for understanding future changes in both water resources and hydrological extremes, such as floods, under climate variability.

## 1 Introduction

Catchment hydrology varies across different time scales: Wetter- and drier-than-usual periods are observed on relatively "short" time scales such as days and seasons (e.g., flood and seasonal regimes) but also on "longer" time scales such as years and decades. For example, southeastern Australia faced a decade-long drought (named the "Millennium Drought") that started in the late 1990s, leading to significant changes in the rainfall–runoff relationship in certain catchments (Fowler *et*

*al.*, 2022). In Europe, different observations have documented flood-rich and flood-poor periods over the past decades and centuries (Blöschl *et al.*, 2020; Wilhelm *et al.*, 2022; Tarasova *et al.*, 2023). Our understanding of such "long-term" variability is still limited compared with the understanding of the daily and seasonal variability (Montanari, 2012), mainly due to the relatively short period of continuous flow recordings. Yet, detecting and understanding the origin of these periods are essential in the context of climate change and for projections of changes in water resources and associated extreme events

such as droughts and floods (Blöschl *et al.*, 2019).

During the past few decades, changes in catchment time series were sought based on different assumptions and therefore different methods and tools. The first common assumption is that linear or monotonous trends may be present in hydrometeorological data. Stahl *et al.* (2010) performed one of the first pan-European analyses looking at streamflow trends in monthly streamflow for the period 1962–2004 across 441 catchments. They highlighted decreasing streamflow trends at





the annual scale in the southern and eastern European regions and positive trends elsewhere. Masseroni *et al.* (2021) recently analyzed trends in the annual streamflow volume from 1950 using a larger set of catchments (more than 3400) and also showed significant negative trends for the Mediterranean catchments and positive trends in the northern regions. These positive trends were also detected by Teutschbein *et al.* (2022) in 50 catchments in Sweden over the past 60 years. Gudmundsson *et al.* (2019) used the GSIM dataset (Do *et al.*, 2018; Gudmundsson *et al.*, 2018) to discuss changes in low,

mean, and high streamflow values at the regional scale, and they identified negative trends in all flow indices for the southern regions of Europe and positive trends in the northern regions. Although Nasreen *et al.* (2022) reported negative trends when analyzing their 500-year annual flow reconstruction over 14 European catchments, the associated signal "is not linear," with wetter and drier periods identified for the catchments studied. These analyses reveal significant hydrological trends at the European scale since the 1950s, with wetter catchments in the north and drier catchments in the south; but these

analyses lead to more nuanced conclusions when viewed from a deeper historical perspective.

Another common assumption is that there are potential periodicities in the variability in hydrological processes over several decades and that these periodicities can be identified using signal-processing techniques. Applying wavelet analysis to more than 1800 monthly streamflow series available since 1962 over western Europe, Lorenzo-Lacruz *et al.* (2022) identified a 7-year cycle in a large proportion of catchments since the mid-1980s. This cycle was not present in earlier periods, suggesting

recent changes in the periodicities of streamflows over the study regions. A 7.5-year periodicity was also identified by Rust *et al.* (2022) when correlating monthly streamflow variations of 767 UK catchments and the North Atlantic Oscillation (NAO). Fossa *et al.* (2021) studied 152 French catchments since 1958 and identified three significant timescales of variability (1, 2–4, and 5–8 years). Thus, these studies reveal significant periodicities of streamflows over particular European regions, in relation to large-scale climate variability and periodicity. For example, Haslinger *et al.* (2021)

highlighted a significant multi-decadal variability in summer precipitation, potentially due to changes in atmospheric circulation related to the Atlantic Multidecadal Variability (AMV), as was previously shown for the northern part of Europe (Ghosh *et al.*, 2017). The multi-decadal variability in floods has been illustrated at the European scale by Blöschl *et al.* (2020) and Brönnimann *et al.* (2022), identifying "flood-rich" and "flood-poor" periods, linked to changes in air temperature, atmospheric circulation, and atmospheric moisture. Renard *et al.* (2023) also identified flood "hot moments"

and "hot spots" at the global scale in their analysis of 180-year flood and heavy precipitation reconstructions. Giuntoli *et al.* (2013) found both a significant increasing trend in drought severity and a correlation with climate indices (e.g., the North Atlantic Oscillation (NAO) and the Atlantic Multidecadal Oscillation (AMO)) in southern France since 1948, and thus stated that *"these observations highlight the difficulties in distinguishing between long-term trends and low-frequency variability based on relatively short series."*

A major challenge in identifying trends, periodicity, or both in catchments at the European scale is how to cope with the variability in length and continuity of hydrometeorological observations. One way to extend the period of observation both in space and time is to use dedicated reanalyses as inputs of rainfall–runoff models. In this context, several global reanalyses such as the NOAA 20CR reanalysis (Slivinski *et al.*, 2019) have been specifically produced for the assessment of the past





century. In its third revision, 20CR is available over the period 1836–2015 and provides eight-times daily meteorological
values across 75-km grids. This reanalysis has been used as boundary conditions for the simulation of an extreme event
(Parodi *et al.*, 2017), to discuss trends in weather patterns over specific regions (e.g., Blanc *et al.*, 2022), or for the
reconstitution of past precipitation using analog methods (Horton, 2022). Yet, such reanalyses are not widely used as inputs
of rainfall–runoff models to study multi-decadal variations in catchment hydrology at the continental scale, being applied at
the national scale instead, such as in France (e.g., Kuentz *et al.*, 2015; Bonnet *et al.*, 2020; Devers *et al.*, 2021). Simulating
catchment hydrology at the multi-decadal scale using such reanalyses faces two main limitations. Firstly, downscaling might
be required to (i) bridge the scale gap between global reanalysis and catchment hydrology and also to (ii) correct long-term
bias potentially present in the reanalysis, e.g., long-term biases in air temperature and precipitation over France were
highlighted by Caillouet *et al.* (2016) and Bonnet *et al.* (2017). Secondly, the use of hydrological models over long and over
hydrologically contrasting periods is limited and associated with uncertainty (e.g., Brigode *et al.*, 2013; Trotter *et al.*, 2023).
Thus, the models need to be calibrated over the past decades (e.g., ~1980–2020) using a reference hydrometeorological
dataset before being used over multi-decadal periods (e.g., Brigode *et al.*, 2016). Despite these two main limitations, such a
modeling approach offers the opportunity to understand, at large spatial and temporal scales, the documented changes in the
past in terms of rainfall–runoff relationships and processes. Moreover, hydrological models are useful to illustrate how
catchments can play a role in amplifying or weakening air temperature and precipitation signals (e.g., Müller *et al.*, 2021;
Baulon *et al.*, 2022; David & Frasson, 2023).

The general objective of this paper is to document the ability of such modeling methodology to identify trends and/or
periodicities of catchment hydrology at the European scale despite the coarse spatial resolution of the global reanalyses and
the rainfall–runoff modeling uncertainty. To this end, we used two global reanalyses (NOAA 20CR and ERA-20C; Poli *et
al.*, 2016) as inputs of a conceptual rainfall–runoff model (GR4J) over 2128 European catchments to simulate daily
streamflows since the 1840s. More specifically, we aimed to answer these three questions:

- How efficient are these two global reanalyses in terms of reconstructing catchment hydrology?
- Does the performance depend on the spatial scale (catchment size) and the hydrological processes studied (catchment regimes)?
- Is the low-frequency variability simulated using this methodology in agreement with observations and other simulation results?

## 2 Data

### 2.1 Climate forcings

Several meteorological databases were assembled for this study. The first objective was to have a reference meteorological
forcing over the recent period (typically the past four decades) that would be homogeneous for all catchments. This
meteorological forcing enables a "classic" calibration of rainfall–runoff models at a spatial resolution that fits the catchment





area. A common forcing set for all catchments studied ruled out the approach of considering the forcings provided in certain CAMELS-type databases. Therefore, two meteorological forcings were extracted over each catchment:

- Catchment daily precipitations were estimated using the MSWEP (V2) dataset (Beck *et al.*, 2018), providing daily precipitation over the period 1979–2019 and a 0.1° (~91 km²) grid.

- Catchment mean daily air temperatures were estimated using the ERA5 reanalysis (Hersbach *et al.*, 2020), providing hourly variables over the period 1980–2019 and a 0.25° (~580 km²) grid.

The combination of MSWEP precipitation time series and ERA5 air temperature time series is denoted as "MSWEP & ERA5" hereafter.

Then, daily precipitation and mean daily air temperature of two long-term historical forcings were extracted over each

catchment studied:

- The ERA-20C reanalysis (Poli *et al.*, 2016), available over the period 1900–2010 with a spatial resolution of 1.4° (~17 000 km²), and denoted as "ERA" hereafter.

- The NOAA 20CR (v3) reanalysis (Slivinski *et al.*, 2019), available over the period 1836–2015 with a spatial resolution of 1.0° (~10 000 km²), and denoted as "NOAA" hereafter.

The common period of the four meteorological forcings is the period 1979–2010 (32 years).

## 2.2 Catchment set

### 2.2.1 Data source and catchment sample selection

An initial sample of 4396 European catchments was assembled from a collection of several "CAMELS-like" datasets (cf. Table 1). This sample groups catchments with daily streamflow series from national datasets (CAMELS-CH (Höge *et al.*,

2023) in Switzerland, CAMELS-FR (Delaigue *et al.*, submitted) in France, CAMELS-GB (Coxon *et al.*, 2020) in the United Kingdom, the NVE (The Norwegian Water Resources and Energy Directorate) dataset (using the NVE Hydrological API (HydAPI), https://hydapi.nve.no/) in Norway, the SMHI dataset (https://www.smhi.se/) in Sweden, and SAIH-RODEA (Yeste *et al.*, 2024) in Spain), transnational datasets (LamaH-CE (Klingler *et al.*, 2021) for Austria, Germany, the Czech Republic, Switzerland, Slovakia, Italy, Liechtenstein, Slovenia and Hungary), or global datasets (GRDC, 2021). Note that 77

duplicate stations were removed from the CAMELS-FR, CAMELS-CH, GRDC, and LamaH-CE datasets.

Four criteria were applied to select a sub-sample of European catchments with (i) a relatively long streamflow time series, (ii) an "adequate" catchment area (relative to the daily time step, the resolution of the climate forcings, the processes studied, and the model used), and (iii) displaying a level of upstream human disturbance as low as possible. Thus, we excluded catchments with:



• Less than 10 years of daily streamflow available over the period 1996–2010 (the period used for the model calibration; see Section 3.2);

• Less than 10 years of daily streamflow available over the period 1982–1995 (the period used for the model evaluation; see Section 3.2);

• Catchment area smaller than 100 km² to avoid issues related to the daily time step and the spatial resolution of
climate forcings;

• Catchment equivalent water depth of storage capacity greater than 10 mm. This equivalent water depth has been estimated, for each catchment, by dividing the total water storage capacity of all dams identified within the catchment using the GRanD dataset (Lehner *et al.*, 2011) by the catchment area (see Delaigue *et al.*, submitted).

After applying these four criteria, the final subset was composed of 2128 catchments.

### 2.2.2 Catchment delineations and hypsometric data

The catchment delineations were extracted from the "CAMELS-like" dataset when available (CAMELS-CH, CAMELS-FR, CAMELS-GB, and LamaH-CE) or were estimated using TauDEM routines (Tarboton, 2013) by positioning manually the hydrometric stations on the theoretical river network estimated using the EU-DEM (v1.1) dataset. This DEM is available at a spatial resolution of 25 m. A comparison between the reference catchment area (i.e., given by the data producer) and the
DEM-derived catchment area was performed to ensure catchment area coherence (not shown here). Hypsometric data were also calculated for each catchment using the EU-DEM dataset.

### 2.2.3 Catchment characteristics

Catchments were grouped by their (i) hydrological regimes and (ii) regions. The catchment regimes were derived using the classification proposed by Hashemi *et al.* (2022) considering interannual monthly catchment air temperature (from ERA5
dataset; see Section 2.1), precipitation (from MSWEP dataset; see Section 2.1), and streamflow over the period 2001–2015. The catchment set is thus composed of (see Fig. 1b):

• 467 nival catchments

• 269 nival-pluvial catchments

• 902 pluvial catchments

• 130 Mediterranean catchments

• 108 uniform catchments





**Table 1: List of catchment datasets used.**

| Country or region | Initial number of catchments | Number of catchments | Reference | Period | Catchment boundaries? |
|---|---|---|---|---|---|
| Switzerland | 211 | 108 | CAMELS-CH (Höge *et al.*, 2023) | 1981–2020 | Y |
| France | 2201 | 783 | CAMELS-FR (Delaigue *et al.*, submitted) | 1900–2021 | Y |
| United Kingdom | 661 | 394 | CAMELS-GB (Coxon *et al.*, 2020) | 1970–2015 | Y |
| Central Europe | 799 | 466 | LamaH-CE (Klingler *et al.*, 2021) | 1981–2017 | Y |
| Norway | 75 | 29 | NVE, https://hydapi.nve.no/ | 1912–2019 | N |
| Sweden | 256 | 192 | SMHI, https://www.smhi.se/ | 1900–2019 | N |
| Europe | 98 | 91 | GRDC (GRDC, 2021) | 1826–2021 | N |
| Spain | 95 | 65 | SAIH-ROEA (Yeste *et al.*, 2024) | 1912–2020 | N |
| **Total** | **4396** | **2128** | | | |

Catchments are also assigned to one of the eight regions inspired by the eight European regions used by Christensen &
Christensen (2007). The catchment set is thus composed of (see Fig. 1c):

- 442 catchments in the Alps

- 395 catchments in the British Isles

- 139 catchments in eastern Europe

- 470 catchments in western France

- 65 catchments in the Iberian Peninsula

- 68 catchments in the Mediterranean region

- 328 catchments in central Europe

- 221 catchments in Scandinavia



Depending on the data source, streamflow time series span over different periods. The streamflow time series starts in 1900

for 25 catchments (1% of the total catchment set), in 1950 for around 200 catchments (10% of the total catchment set), and

in 1970 for half of the catchment set (Fig. 1d).



**Figure 1: (a) Source, (b) hydrological regimes, (c) regions, and (d) temporal availability of daily streamflow data of the study catchments.**

## 3 Methods

### 3.1 Rainfall–runoff model

The GR4J (Perrin *et al.*, 2003) conceptual rainfall–runoff model and its snow accumulation and melting module CemaNeige (Valéry *et al.*, 2014) were used for streamflow simulations. The inputs of this model are:





- Daily precipitation (in mm/j)

- Daily potential evapotranspiration (in mm/j), estimated using the Oudin *et al.* (2005) formulation, considering daily air temperature and mean catchment latitude.

- Daily air temperature (°C).

## 3.2 Model parameter calibration

The GR4J and CemaNeige models have, respectively, four and two free parameters that need to be calibrated conjointly for
each catchment, considering the observed daily streamflow available over a given time period. The parameter calibration was performed individually for each catchment over a common period comprising 10–15 years over the period 1996–2010, with a warm-up period of 3 years from 1993 to 1995. The objective function used for the model calibration is the Kling and Gupta efficiency criterion (KGE; Gupta *et al.*, 2009), ranging from -∞ to 1 and estimated as follows:

$$KGE = 1 - \sqrt{(\beta - 1)^2 + (\alpha - 1)^2 + (r - 1)^2} \tag{1}$$

where:

- $\beta$ is the ratio between the means of the simulated and observed time series; this quantifies the simulation bias, and ranges between 0 and +∞ (values > 1 indicate a model overestimation).

- $\alpha$ is the ratio between the standard deviations of the simulated and observed time series; this quantifies the ability of the simulation to reproduce the streamflow variability, and ranges between 0 and +∞ (values > 1 indicate a model
overdispersion).

- $r$ is the coefficient of correlation between the simulated and the observed series; this quantifies the ability of the simulation to reproduce the observed temporal variations, and ranges between -1 and 1 (perfect correlation).

Three different meteorological forcings were used for the model parameter calibration, thus resulting in three different parameter sets for each catchment:

- Parameter sets calibrated considering precipitation and air temperature from MSWEP & ERA5,

- Parameter sets calibrated considering precipitation and air temperature from ERA,

- Parameter sets calibrated considering precipitation and air temperature from NOAA.

The model was manipulated using the R (R Core Team, 2020) package airGR (Coron *et al.*, 2017, 2023).



## 3.3 Model evaluation

### 3.3.1 Periods of daily streamflow evaluation

The daily streamflow simulations were evaluated over two different time periods (Fig. 2):

1. To compare the three meteorological forcings (MSWEP & ERA5, NOAA, and ERA), the period 1982–1995 (14 years) was considered, with 3 years for model warm-up (1979–1981).

2. A longer period was considered to compare the two historical meteorological forcings (NOAA and ERA) by using all available years from 1903 to 1995, with 3 years for model warm-up (1900–1902).

The evaluation metrics are the KGE values and its three components (β, α, and $r$; see Equation 1).

### 3.3.2 Annual time series at the catchment scale

For each catchment, the daily times series (observations, NOAA simulations, and ERA simulations) are aggregated at the annual time step for the specific evaluation of:

- Mean flow: calculation for each year of the mean annual streamflow, named "$Q_A$" hereafter

- High flow: calculation for each year of the maximum streamflow value, named "$Q_X$" hereafter

- Low flow: calculation for each year of the minimum mean monthly streamflow value, named "$Q_M$" hereafter.

To assess the ability of the model to reproduce the long-term temporal variability of the streamflow time series, temporal correlations between simulations and observations were estimated for each variable ($Q_A$, $Q_X$, and $Q_M$) for each catchment individually:

- On the annual time series

- On 10-year running mean time series

In each case, correlations are estimated only if the observed series length exceeds 30 years.

### 3.3.1 Annual time series at the regional scale

Finally, regional anomalies of mean, high, and low flows were calculated. For each region and each year, catchments with data were identified: If more than 10 catchments are available for a given year and region, an anomaly is calculated by dividing the annual flow values by the average of the annual flow values for the catchment subset studied. Thus, the catchment subset may change every year. Finally, the 10-year running mean is calculated for mean-, high-, and low-flow indices.





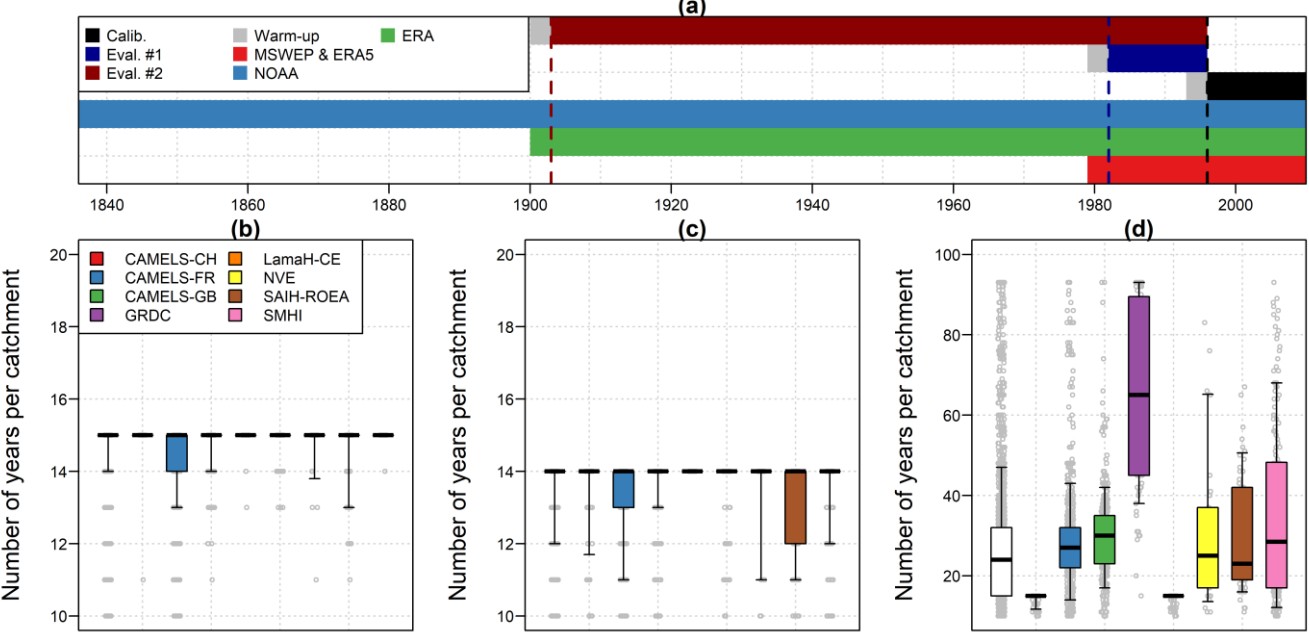

**Figure 2: (a) Temporal availability of meteorological forcings and number of streamflow observations per catchment over (b) the calibration period (1996–2010), (c) evaluation period 1 (1982–1995), and (d) evaluation period 2 (1903–1995).**

## 4 Results

### 4.1 Model calibration performance

Figure 3 presents the rainfall–runoff model performance over the calibration period for the three different climate forcings (MSWEP & ERA5, NOAA, and ERA). The calibration performance obtained using the MSWEP & ERA5 meteorological forcings is relatively good, with a slightly worse performance for catchments in eastern Europe, the Iberian Peninsula, and Mediterranean regions. The performance obtained using NOAA and ERA forcings is somewhat lower, being average to poor. The rainfall–runoff model achieves better performance with NOAA precipitation and temperature compared with

ERA. Nevertheless, the performance of the model in each region is similar depending on the meteorological dataset used, with the worst performance encountered for catchments in eastern Europe and Mediterranean regions.





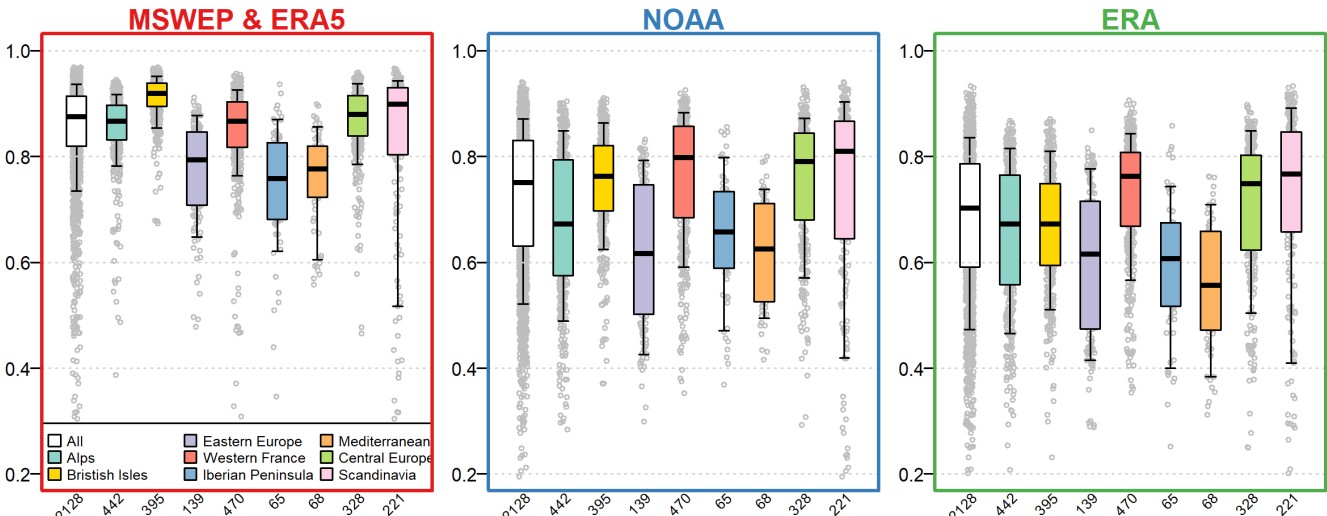

**Figure 3: Calibration performance (KGE) for three different climate forcings (MSWEP & ERA5, NOAA, and ERA, from left to right) according to region. Boxplots are constructed with the 0.10, 0.25, 0.50, 0.75, and 0.90 quantiles.**

## 4.2 Evaluation performance

### 4.2.1 Daily streamflow simulations

Figure 4 presents the rainfall–runoff model performance over the calibration period and first evaluation period, grouped according to the parameter sets used (calibrated using MSWEP & ERA5, NOAA, or ERA forcings over the calibration period) and according to the climate forcing used (NOAA or ERA). The performance obtained using parameters calibrated with MSWEP & ERA5 forcings is poor when using both NOAA and ERA forcings. Logically, when using NOAA (ERA) forcings, the performance obtained using parameters calibrated with NOAA (ERA) forcings is better than using parameter sets obtained with ERA (NOAA) forcings. Finally, the performance obtained over the two evaluation periods is higher when using NOAA forcings than ERA forcings ($p$ value of Wilcoxon rank test (1945) < 2.2e-16 when comparing performance using NOAA parameter sets and climate forcings with performance using ERA parameter sets and climate forcing).


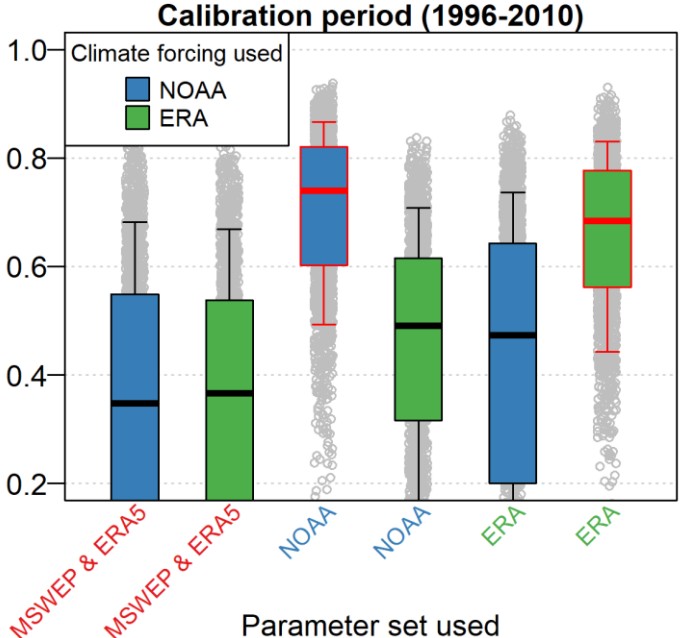

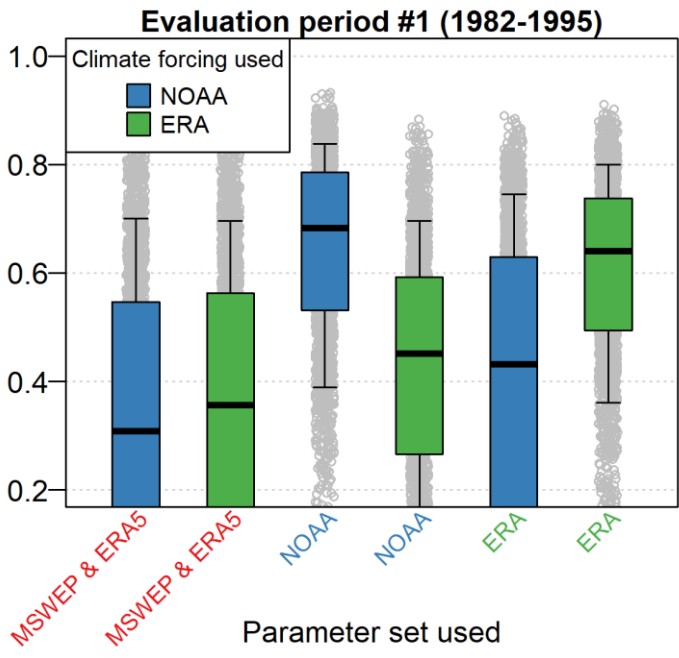


**Figure 4: Evaluation performance (KGE) over the calibration period (top) and evaluation period 1 (bottom), grouped by the parameter sets considered and the climate forcing used. Boxplots are constructed with the 0.10, 0.25, 0.50, 0.75, and 0.90 quantiles. The boxplots framed in red summarize the performance obtained in calibration.**





Figure 5 presents the KGE and KGE components calculated over the two evaluation periods, considering each forcing for
parameter calibration and as meteorological forcings. This figure shows that NOAA simulations are more closely correlated
with observations (*r*) compared with ERA simulations: (*p* value of Wilcoxon rank test (1945) < 2.2e-16 when comparing *r*
values obtained using NOAA parameter sets and climate forcings with *r* values obtained using ERA parameter sets and
climate forcings). Mean bias (beta) and deviation bias (alpha) reveal an overall slight underestimation of streamflow values
and variance by ERA simulations.

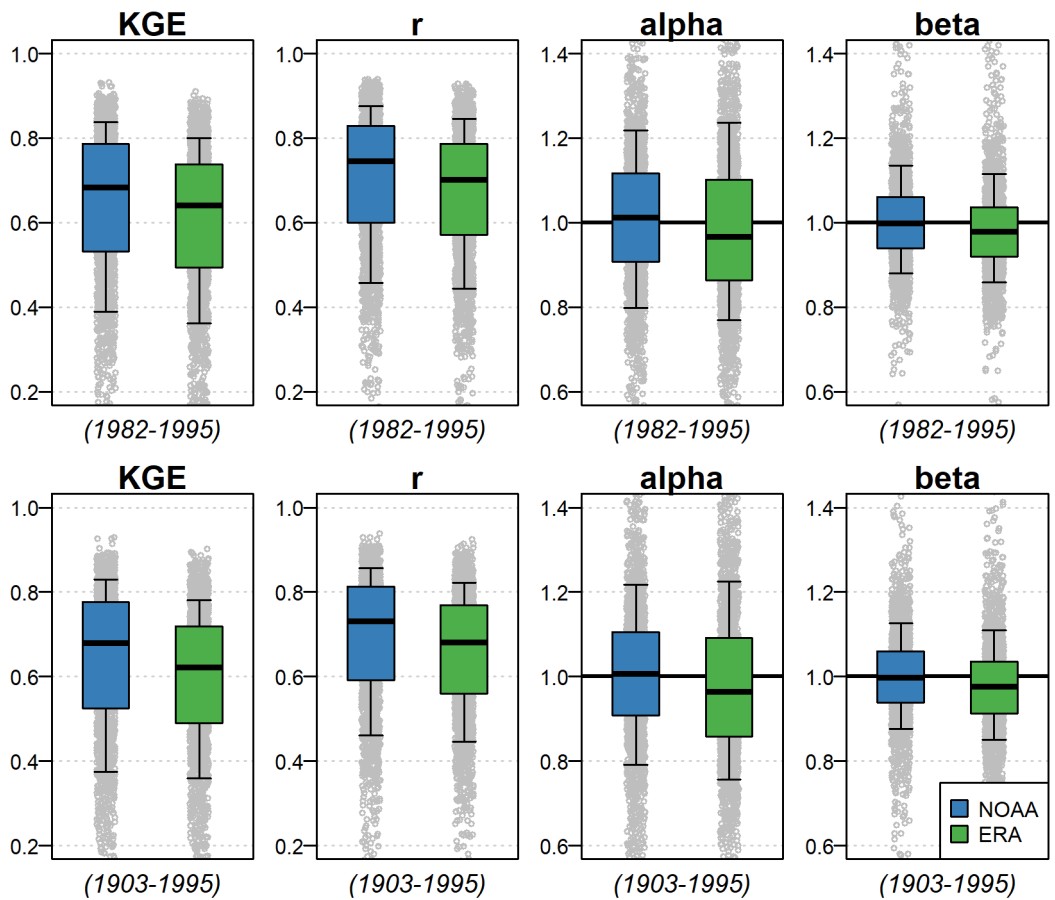


**Figure 5: Evaluation performance (KGE and KGE components) over the two evaluation periods (1982–1995 and 1903–1995, from top to bottom), with catchments grouped by the climate forcing used for parameter calibration and simulation. Boxplots are constructed with the 0.10, 0.25, 0.50, 0.75, and 0.90 quantiles.**

### 4.2.2 Simulations aggregated over time

Figure 6 shows the temporal correlations evaluated individually for each catchment over the 1903–1995 evaluation period
for the daily simulations aggregated at the annual time step. The overall performance is better for mean flows than for low
flows and high flows. The difference between the two simulation types (NOAA or ERA) is not clear, with NOAA
simulations being marginally better and in relative agreement when analyzed over the geographical regions. The





performance is, for mean, low, and high flows, dependent on the region: For mean flows, the performance obtained over
regions in western France is clearly the best, while the performance obtained over the Alps, eastern Europe, and the Iberian
Peninsula is the worst. For low and high flows, the performance over the Alps, eastern Europe, and the Iberian Peninsula is
also the worst for NOAA simulations. For ERA, the low-flow simulations are the worst for the Iberian Peninsula, while the
high-flow simulations are the worst for the Alps, eastern Europe, Iberian Peninsula, and Mediterranean regions. For both
NOAA and ERA, the best performance for low and high flow is found in western France, central Europe, and Scandinavian
regions. These general conclusions are also valid when looking at the flow annual time series smoothed with a 10-year time
step (not shown here).

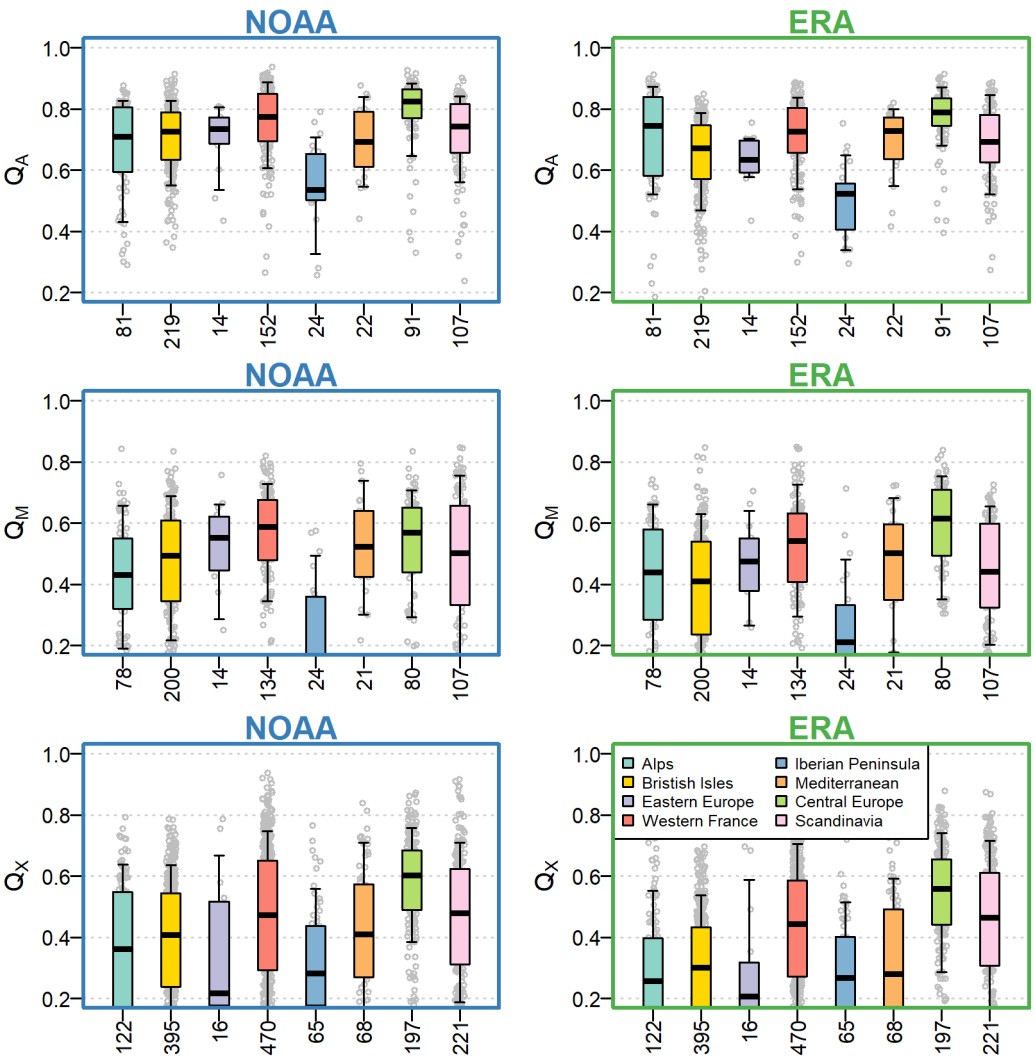

**Figure 6: Temporal correlation (r) over the 1903–1995 evaluation period, with catchments grouped by the climate forcing used (NOAA or ERA) and the catchment region. Boxplots are constructed with the 0.10, 0.25, 0.50, 0.75, and 0.90 quantiles. The x axis**
**shows the number of catchments available for each region.**



### 4.2.3 Simulations aggregated over time and space

Figure 7 compares the annual anomalies of mean flow ($Q_A$) observations and simulations for each region studied. The two climate forcings reproduce relatively well the interannual variability in regional mean flows, with a good correlation between observed and simulated successions of wetter/drier sub-periods. No clear trend emerges in terms of dependence on the region
or time period considered. Unfortunately, this analysis is limited for regions with few data available (e.g., eastern Europe and the Mediterranean region). Similar conclusions are reached when analyzing low and high flows (cf. Fig. 8 and Fig. 9), albeit with a higher amplitude of anomalies in observations and simulations.

**Figure 7: 10-year running means of the QA observations (black) and simulations (blue: NOAA simulations, green: ERA
simulations) for each region studied. Only years with at least 10 available catchments per region are considered. The red y axis shows the number of available catchments per year and region.**



**Figure 8: 10-year running means of the QM observations (black) and simulations (blue: NOAA simulations, green: ERA simulations) for each region studied. Only years with at least 10 available catchments per region are considered. The red y axis shows the number of available catchments per year and region.**







**Figure 9: 10-year running means of the QX observations (black) and simulations (blue: NOAA simulations, green: ERA simulations) for each region studied. Only years with at least 10 available catchments per region are considered. The red y axis shows the number of available catchments per year and region.**

## 5 Discussion

### 5.1 Method and data used for the multi-decadal hydrological simulation

In comparing the performance obtained depending on the forcings and parameter sets used, we aimed to show that it was more appropriate to use forcings with the finest spatial resolution (in this case, MSWEP and ERA5 forcings) to calibrate the model parameters. However, this hypothesis proved to be false: When NOAA forcings are used as input to the model, the parameter sets obtained after calibration with MSWEP & ERA5 perform worse than those obtained after calibration with ERA, which in turn are less effective than those obtained after calibration with NOAA. Thus, consistency between the forcings used in calibration and simulation appears to be more important than the spatial resolution of the forcings used during calibration. This result can be explained by the flexibility of the rainfall–runoff model during parameter calibration, allowing for an implicit adaptation of the model parameters to the spatial resolution of the forcings. In our study, the





calibration period was temporally restricted to be common across the different forcings. Nevertheless, this result advocates extending the calibration period to cover the entire period of available discharge data for each catchment.

The simulation methodology used in this study has several important limitations to be noted. First, it is based on a single
conceptual rainfall–runoff model. Using a multi-model approach would make it possible to quantify the uncertainty associated with the model structure in the simulations performed (Wan *et al.*, 2021; Martel *et al.*, 2023; Thébault *et al.*, 2024). The simulation method was applied at a daily time step, which may be too coarse for flood modeling in some small and/or Mediterranean catchments. The Oudin *et al.* (2005) formula used to estimate potential evapotranspiration series at the catchment scale is also a significant source of uncertainty in the context of streamflow simulation (Lemaitre-Basset *et al.*,
2022a). The use of different formulations, for example, taking into account $CO_2$ (Lemaitre-Basset *et al.*, 2022b) is an interesting perspective of this work.

The objective function used in this study is also an important methodological choice, and could have been adapted to the flows studied (e.g., by using an objective function better suited to reproducing low flows; Pushpalatha *et al.*, 2012) and also adapted to the modeling exercise over a long period (e.g., Split KGE proposed by Fowler *et al.* (2018) for the simulation of
drying climate in Australia).

Furthermore, the rainfall–runoff model parameters were estimated over a short and recent period relative to the entire period considered for streamflow reconstruction. Many studies have highlighted the significant difficulty hydrological models face in simulating periods with different climatic conditions than those considered for parameter calibration (e.g., by Fowler *et al.*, 2020; Duethmann *et al.*, 2020). This uncertainty could be quantified by considering, for each catchment, several
parameter sets per hydrological model (e.g., after calibrations over different sub-periods through bootstrapping of the observations, Brigode *et al.*, 2015; Arsenault *et al.*, 2018).

Finally, the selection of catchments for this analysis raises some important points. Certain regions are over-represented while others are under-represented, and the uncertainty associated with the hydrometric data has not been addressed. Furthermore, the analyses performed to assess the "natural" characteristics of the catchments—those with minimal anthropogenic
influences such as significant water withdrawals, dams, or wastewater treatment plants—were limited and did not account for the temporal evolution of these influences over the periods studied. Additionally, the availability of long streamflow time series (i.e., spanning several decades) is scarce, highlighting the need to explore the possibility of conducting this analysis with a dataset constructed at a monthly time scale.

**5.2 Variability in performance within the catchment set**

The analysis of the performance obtained during the calibration of the rainfall–runoff model showed poorer results for Mediterranean catchments. This poor performance can be explained by various factors, such as the spatial resolutions of the meteorological forcings used (greater or equal to 10,000 km²), which may be too coarse relative to the size of the catchment to capture the relevant processes. Other factors include significant anthropogenic influences that may vary over time in the





streamflow observations used or the model's difficulty in simulating the hydrological regime intermittency. The performance

on eastern European basins is also weaker than for other regions; thus, further investigations are needed to explain this trend.

Figure 10 shows the relation between the evaluation performance at the daily time step and the catchment regimes and areas. This figure reveals an increase in performance with the size of the catchment, up to a threshold beyond which performance no longer improves. This conclusion does not hold true for Mediterranean catchments and is less true for snow-dominated catchments.

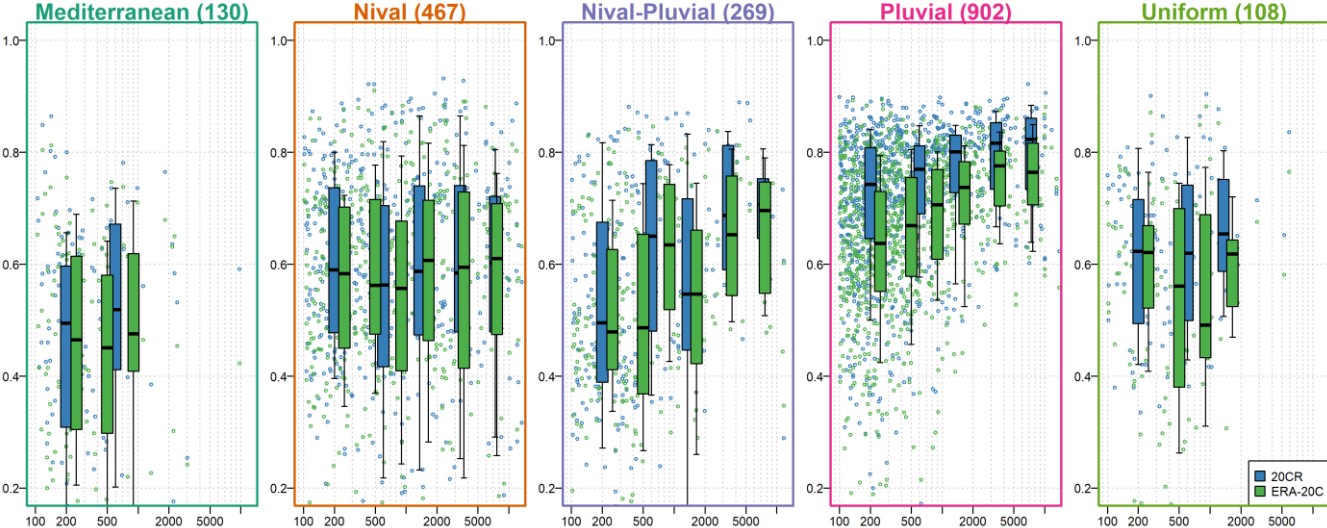


**Figure 10: Evaluation performance (KGE) over evaluation period 1 grouped by catchment regime and area (in km²). Boxplots are constructed with the 0.10, 0.25, 0.50, 0.75, and 0.90 quantiles.**

As expected, the simulation performance is better for annual mean flows than for minimum monthly flows and annual maximum daily flows, for both forcings tested. This result can probably be explained by (i) the method used for the

hydrological simulation (discussed in Section 5.1) and more specifically the choice of the objective function used to calibrate the model (which can be adapted to a particular objective) and (ii) the quality of the climatic forcings used, characterized by a spatiotemporal resolution that is probably too coarse to represent the variability in the hydrological processes that generate floods.

Figure 11 shows the performance of high-, mean-, and low-flow simulations smoothed over 10 years as a function of the

performance of simulations of daily streamflows. For the two climate forcings and the three annual flows, no clear relationship is identified, showing an independence between performance at the daily time step and the model's ability to represent interannual flow variability.





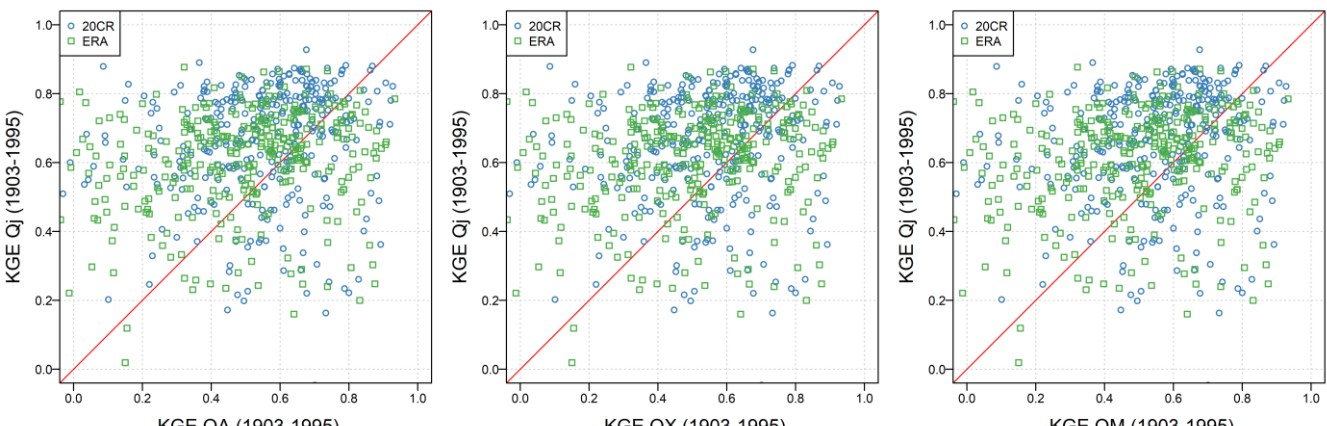

**Figure 11: Comparison between simulation performance of mean- (left), high- (center), and low-flow (right) simulations (smoothed over 10 years) and the performance of simulations of daily streamflows.**

### 5.3 Consistency of multi-decadal hydrological variability

The interannual variability simulated using the two climate forcings (20CR and ERA) shows reasonable consistency with available observed data at the regional scale. When analyzing 10-year running means, the temporal correlations reach approximately 0.8 for mean flows and 0.6 for both low and high flows (Fig. 12). On average, the 20CR reanalysis performs slightly better than the ERA reanalysis across the three flows studied. However, when zooming in at the regional scale, this ranking is reversed for the central European and Mediterranean regions for mean and low flows, as well as for the eastern European region in terms of high flows. The simulation performance for high flows is notably poor for both reanalyses in the Alps and eastern European regions. It is important to note that the analysis of consistency between simulations and observations was conducted over large, somewhat arbitrary regions, which may group together catchments with different hydrological regimes. A similar analysis could be conducted at a finer spatial scale using more precisely "hydrologically defined" regions.

Reviews of hydrological trends in Europe since the 1950s suggest a general tendency for northern European catchments to become wetter, while southern European catchments tend to dry out (e.g., Masseroni et al., 2021). This drying trend is evident in both the observations and the simulations for the Mediterranean and Iberian Peninsula catchments, affecting mean, low, and high flows. Although the past few decades appear to have been relatively wetter for the Scandinavian catchment studied, no clear linear trends are observed for these basins.

In general, the analysis reveals alternating dry and wet periods across all regions and flow indicators studied, rather than consistent linear trends. It is noteworthy that within the same region, individual analyses of interannual variations for low, mean, and high flows can reveal different anomalies. For instance, in the British Isles, the years 1980 to 1985 appear relatively wet across low, mean, and high flows, while the period from 1940 to 1950 is only characterized as wet in terms of high flows. Similarly, for the central European catchments, there was a dry period for high flows between 1930 and 1935,





followed by a wet period between 1940 and 1945. This anomaly is not reflected in the analysis of low flows, highlighting differing interannual variations between flow types. Most of these trends have already been identified in the literature. For instance, the wet periods (1920s, 1980–1990) and the dry decade (1970s) observed by Lindström & Bergström (2004) in

Sweden, the multi-decadal variations simulated with a similar method by Devers *et al.* (2024) in France, or the "flood-poor" period identified after World War II by Brönnimann *et al.* (2022) at the European scale.

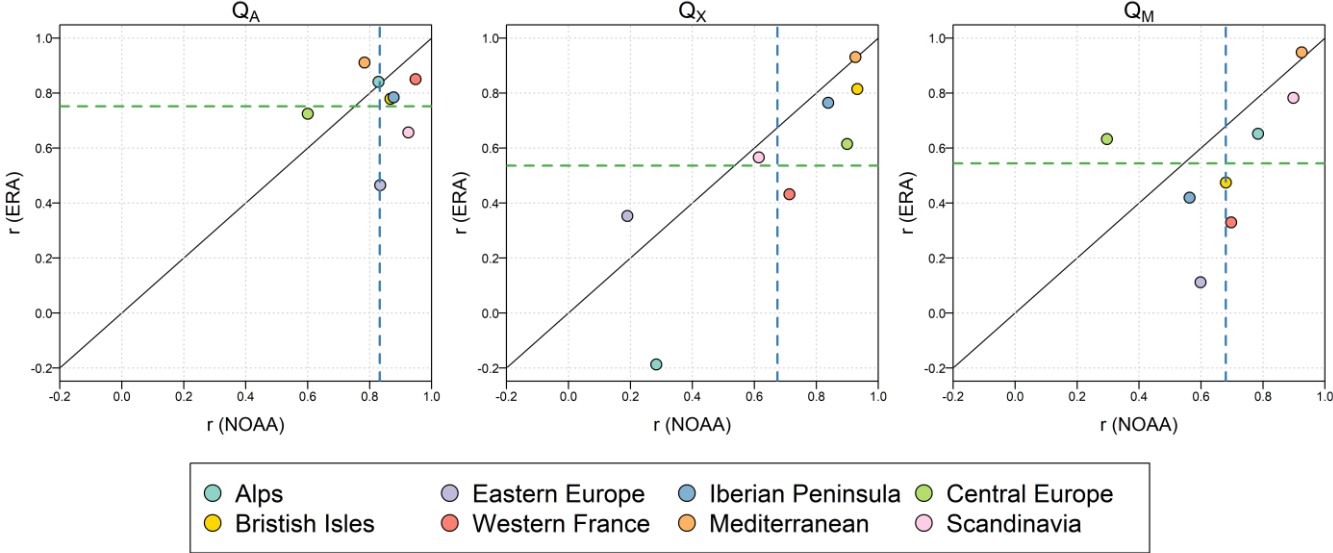

**Figure 12: Temporal correlation (r) over the 1903–1995 evaluation period between 10-year running means of the QA, QX, and QM observations and simulations for each region studied. Only years with at least 10 available catchments per region are**

**considered. Dashed blue (resp. green) line represents the average correlation coefficient over all regions obtained using NOAA (resp. ERA).**

## 6 Conclusions

The objective of this study was to quantify the ability of global reanalysis used as input of a conceptual rainfall–runoff model to simulate catchment hydrology at the European scale. Thus, we used two global reanalyses (NOAA 20CR and ERA-20C)

as inputs of the GR4J conceptual rainfall–runoff model over 2128 European catchments to simulate daily streamflows since the 1840s. The results obtained enable us to provide some answers to the three questions that interested us specifically:

- **How efficient are these two global reanalyses in terms of reconstructing catchment hydrology?**

The two reanalyses tested here yielded a relatively good performance across all the catchments studied, both at the daily time step and for decadal flow anomalies. The performance is better for mean flows than for low and high flows. An important

result to be highlighted is the necessary consistency between the forcings used for calibration and those used for simulation: The best simulation performance with the ERA-20C forcing was achieved when using parameters calibrated with the same



forcing, even though it is coarser than the other forcings considered. Thus, consistency between the forcings used in the calibration and simulation appears to be more important than the spatial resolution of the forcings used during calibration.

- **Does the performance depend on the spatial scale (catchment size) and the hydrological processes studied (catchment regimes)?**

In general, the performance increases with catchment size (except for Mediterranean and snow-dominated catchments), up to a threshold beyond which no further improvement is observed. Additionally, the performance varies across regions, reflecting the different hydrological processes in the catchments studied. The best performance is typically seen in catchments located in western France, Scandinavia, and the British Isles, while the lowest performance is observed in eastern Europe, the Mediterranean, and the Iberian Peninsula. Therefore, the ability of these two climate forcings and the modeling chain used to represent processes related to snow accumulation and melting, as well as the intermittency of the hydrological cycle, requires further investigation. Finally, the differences between the two reanalyses in certain regions are noteworthy. For example, the temporal correlation on 10-year running means for mean flows in eastern Europe is significantly higher for NOAA 20CR (~0.8) than for ERA-20C (~0.4), suggesting the potential for coupling the two reanalyses in specific regions.

- **Is the low-frequency variability simulated using this methodology in agreement with observations and other simulation results?**

The interannual variability simulated by NOAA 20CR and ERA-20C reanalyses shows good consistency with observed data. The analysis highlights alternating wet and dry periods across all regions, with different anomalies depending on flows, and suggests that the tested methodology holds promise for investigating mechanisms behind these variations to understand regional hydrological changes better. There are opportunities to refine the analysis of these consistencies by comparing it with databases focused on specific flow types, such as the flood dates compiled in the HANZE database (Paprotny *et al.*, 2023).

Finally, this analysis highlights the significant multi-decadal variability in catchment streamflow, which may be underestimated when linear trends are sought in time series spanning only a few decades. In this context, the use of long-term climatological and hydrological reanalyses is crucial, particularly for anticipating the effects of climate change on hydrosystems and for attributing changes at the catchment scale.

**Code availability**

The R (R Core Team, 2020) package airGR (Coron *et al.*, 2017, 2023) was used to perform all hydrological simulations. The R scripts required to use this package in this context can be provided upon request from the corresponding author.



**Data availability**

The climate forcings used in this article can be downloaded online. The streamflow data for the studied catchments were obtained from the open-access databases listed in Table 1 or downloaded online via the APIs described in Table 1.

**Author contribution**

PB conceptualized the work. PB wrote the computer codes to format the data, simulate streamflow from climate forcings and quantify model performance. PB and OD drafted the manuscript. All authors reviewed and edited the manuscript.

**Competing interests**

The authors declare that they have no conflict of interest.

**Acknowledgements**

This work was supported by the French National program EC2CO (Ecosphère Continentale et Côtière)

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
