# Peer review of "Using century-long reanalysis and a rainfall-runoff model to explore multi-decadal variability in catchment hydrology at the European scale"

_Hydrology and Earth System Sciences, 2024_

## Author Response (AR1)

**Reviewer 1**

This paper presents an analysis of the century-long ERA-20C and NOAA-20CR reanalysis products to simulate flows and long-term trends that could otherwise be missed when looking at shorter time periods. I think the study is of interest and proposes an alternative and convincing argument that the streamflows alternate between wet and dry periods over long (i.e. decadal or longer) time periods. While I found the paper to be well written and interesting, I also have some issues that I think would need to be solved before recommending acceptance for publication.

Thank you for this positive feedback on our work.

1. The title should reflect the fact that the reanalyses are not just global reanalyses. They are century-long reanalyses, which is the real kicker and novelty.

Thank you for this suggestion. We changed the title of the article to emphasize the length of the reanalyses used and to incorporate the word "explore" (as suggested by Reviewer 2).

The previous title of the article was: "Using global reanalysis and rainfall-runoff model to study multi-decadal variability in catchment hydrology at the European scale."

We propose the following revised title: "Using century-long reanalysis and a rainfall-runoff model to explore multi-decadal variability in catchment hydrology at the European scale."

2. Line 63: Define what the 20CR and 20C mean (20th century reanalysis, please add)

We clarified the meaning of 20CR in the revised version of the article:

In this context, several global reanalyses such as the NOAA 20CR (20th Century Reanalysis, Slivinski et al., 2019) have been specifically produced for the assessment of the past century.

3. Line 64: "eight-times daily" --> Better to say 3-hourly.

Thank you, we changed this term in the revised version of the article:

In its third revision, 20CR is available over the period 1836–2015 and provides 3-hourly meteorological values across 75-km grids.

4. Line 81: "such modeling" --> such a modelling"

Thank you, we corrected this in the revised version of the article.

5. I see the resolution is a key issue in this paper. For starters, the MSWEP+ERA5 combination is mismatched, and the authors should have considered ERA5-Land to match the MSWEP precipitation resolution (0.1° each). Furthermore, to preserve coherence, ERA5-Land precipitation could have been used instead of MSWEP. I think a justification of this should be

provided as well as an impact analysis (not redoing runs, but perhaps contextualizing with respect to the size of the catchments?).

Thank you for this comment. We agree with the reviewer: the spatial resolution of the input data for the rainfall-runoff model is a key aspect of our article.

For preliminary tests, in addition to the historical reanalyses ERA-20C and NOAA 20CR, we considered two reference datasets for the recent period: (i) the combination of MSWEP precipitation and ERA5 air temperature and (ii) the combination of ERA5-Land precipitation and ERA5 air temperature. The calibration performances obtained with these four datasets are presented in the following figure 1. The results show that the performances obtained using the combination of ERA5-Land precipitation and ERA5 air temperature (in black, not shown in the manuscript) are lower than those obtained with the combination of MSWEP precipitation and ERA5 air temperature. For this reason, we chose not to present the results obtained with the ERA5-Land precipitation and ERA5 air temperature combination in our article. However, we did not test the combination of ERA5-Land precipitation and ERA5-Land air temperature, but it is likely that the worse performance of the ERA5 forcing is due to precipitation data, not temperature data.

Figure 1: Calibration performance (KGE) for four different climate forcings (MSWEP & ERA5, ERA5-Land & ERA5, NOAA, and ERA, from left to right) according to region (note that Spanish catchments are not present in this subset). Boxplots are constructed with the 0.10, 0.25, 0.50, 0.75, and 0.90 quantiles.

Due to the relative length of the paper, we chose not to include this figure, but to mention this preliminary test in Section 2.1 (Climate forcings):

It is worth noting that we also considered an alternative reference dataset, namely ERA5-Land precipitation and ERA5 air temperature. However, since the calibration performance obtained with this forcing was lower than that achieved with the MSWEP & ERA5 dataset (results not shown in this paper), we did not retain this forcing for the remainder of the study.

6. In a similar vein, catchments sizes were restricted to above 100km2, whereas the ERA and NOAA datasets cover swaths of land between 10000 and 17000km2, which is a huge mismatch, especially the strong elevation gradients all over large parts of Europe. Was downscaling not an option? It seems that at least with the altitude/elevation and some background information it would be possible to do at least a rudimentary approach. I think the authors should consider this in their next version, or at least discuss it in more details because it is a key element of the paper.

(This is the same response as the one provided for point 6 raised by Reviewer 2)

We thank the reviewer for this important comment. Yes, we conducted tests to extrapolate meteorological forcings before model parameter calibration and rainfall-runoff simulations, particularly for high-altitude catchments. These tests aimed to extrapolate precipitation and air temperature based on the difference between the median elevation of each catchment and the median elevation of the meteorological forcing grid cells in each dataset. The results showed an improvement in calibration performance for small mountainous catchments but also led to a decrease in performance for other catchments. Since the performance improvement was not consistent across all catchments, we did not pursue this option further in the article and used the meteorological forcings as they were, without downscaling. By doing so, we assume having lower performance in terms of hydrological simulations at the daily time step, but we maintain the long-term trends in air temperature and precipitation from the considered reanalyses.

However, it is important to note that the use of the snow accumulation and melt model CemaNeige inherently involves a downscaling of meteorological forcings by distributing precipitation and air temperature over five elevation bands (in our case), which are constructed based on the hypsometric curve of the catchment. Thus, the forcings input into the model are considered representative of those observed at the median elevation of the catchment and are then distributed across the five zones according to the gradients described by Valéry et al. (2014).

The development of a downscaling method at the scale of all the European catchments used in our study was beyond the scope of our article but represents a natural perspective for future work.

We have added elements to the paper regarding the extrapolation performed using the CemaNeige module in Section 3.1 (rainfall-runoff model):

The CemaNeige module takes into account the hypsometric curve of each catchment to perform a downscaling of meteorological forcings by distributing daily precipitation and air temperatures across five (in our case) elevation bands. Thus, the forcings input into the model are considered representative of the median elevation of the catchment and are then distributed across the five zones according to the gradients described by Valéry et al. (2014).

7. KGE version used is the 2009 version, whereas the community has moved on to the 2012 modified KGE. While I have no problem with this (it is still a good metric), it would be good to explain that this was an editorial choice.

Thank you for this comment. We used the 2009 version of the KGE to be able to compare the performance and parameter values obtained in other similar studies on several catchment subsets (comparisons not shown in the article). We clarified this point in the revised version of the article:

We used the 2009 version of the KGE in order to allow comparison of model performance and parameters with other similar studies conducted on sub-samples of European catchments (comparison not shown in this paper).

8. Line 198: "manipulated" has a negative connotation. I would suggest: "was implemented".

Thank you, we changed this sentence in the revised version of the article:

The model was implemented manipulated using the R (R Core Team, 2020) package airGR (Coron et al., 2017, 2023), using the default optimization algorithm included in the airGR package. This algorithm was specifically designed for GR models (Coron et al., 2017).

9. It seems Figure 2a-c first boxplot is not colored in red as supposed to? Or is it some other variable, given there are 9 boxes and only 8 legend entries? Which seems true for the two other boxplots as well. Please clarify.

Thank you for this comment: the first boxplot in each figure represents all the studied catchments. We clarified this point in the revised version of the figure.

10. Lines 248 and 256: p-value can be set to 0 when it is basically machine precision (2.2e-16).

Thank you, we thresholded the p-value at 0.

11. Figure 5: One problem here is that the top rows will be a significant subset of the bottom rows, so the results are not necessarily comparable. For example, imagine that 50% of the catchments only have validation data on the exact 1982-1995 period. That would mean that those basis' scores would be exactly the same in the bottom plots even though it should cover a longer period, but it cannot be interpreted that way. I would suggest identifying a series of catchments that have at least 50 years of validation data and keeping those independent for the long-duration tests.

Thank you for this suggestion: we present, in the revised version of the article, the results of a subsample of 187 catchments with at least 50 years of data over the second evaluation period:

Figure 5 presents the KGE and KGE components calculated over the two evaluation periods, considering each forcing for parameter calibration and as meteorological forcings. For the second evaluation period (1903-1995), only the 187 catchments with more than 50 years of observations are considered.

12. Lines 275-276: I think this is useful information that should be shown as it would show another mode/dimension to the problem that filters out random perturbations and focuses on the longer-term patterns.

Thank you for this suggestion: we added these additional results in the Appendix A of the revised version of the article to avoid making the article too lengthy.

13. General comment: It would be good to have a series of simulations for which only the ones that obtained KGE in calibration/validation above a certain quality threshold are preserved. Ex KGE > 0.7 for NOAA/NOAA and ERA/ERA (not really useful to do ERA5+MESWEP/NOAA(ERA). This would ensure results are not negatively affected by poorly modelled basins/models, as we have some that have quite low KGE values that contribute to the detailed variability results, and are perhaps not as trustworthy.

Following the reviewer's suggestion, a complementary analysis was performed by retaining only the catchments with calibration/validation KGE values above 0.7 for the NOAA/NOAA and ERA/ERA simulations. The figures corresponding to this subset (equivalent to Figures 6 to 8 in the manuscript) show very similar spatial patterns and variability ranges to those already presented. Therefore, to avoid overloading the article, these additional figures were not included in the revised version. However, this sensitivity test confirms that the conclusions drawn are not significantly influenced by poorly modelled catchments.

14. Figure 6: Not clear why the number of catchments changes for each metric. I would assume they would be the same from one metric to the next since they are only excluded if they don't have 30 years of observations?

Thank you for this comment. The x-axis of Figure 6 was incorrect in the previous version of the manuscript; this error has been corrected in the revised version. The results presented remain unchanged.

15. Figure 6: I would show the boxplots in their entirety here. Not much use limiting to 0.2. At least to 0.0.

Thank you for this comment: we changed the y-axis limit of Figure 6 (and Figure A) in the revised version of the article.

16. Lines 310-311: There have been numerous studies on this previously, I think it would be good to reference a few to show that your results are in-line with the current literature.

Thank you for this comment. We agree with the reviewer and added two references on this topic in the revised version of the article.

Hydrological models have the ability to compensate for errors in forcing via parameter calibration, whether these errors are systematic (bias) or random (see e.g. Dawdy and Bergmann, 1969; Oudin et al., 2006).

17. Line 321: "of this work" --> "for this work"

Thank you, we changed this sentence in the revised version of the article.

18. Lines 334-335: But also human intervention, forestry, agriculture, urbanization over the past ~120 years has definitely had an impact on hydrological response.

We agree with the reviewer on this point and have therefore revised this part of the discussion accordingly:

Furthermore, the identification of catchments with minimal anthropogenic influence relied on a limited set of indicators—such as major water withdrawals, dams, or wastewater treatment plants—and did not account for the uncertainty associated with these data. Other long-term changes likely to affect hydrological response, such as land use changes related to forestry, agriculture, or urbanization over the past century, were not considered in this context.

19. Figure 11: I think there is a problem here, all three figures are exactly the same.

We have corrected Figure 11. The conclusions drawn from the analysis of this figure remain unchanged:

Figure 11 shows the performance of high-, mean-, and low-flow simulations smoothed over 10 years as a function of the performance of simulations of daily streamflows. For the two climate forcings and the three annual flows, no clear relationship is identified, showing an independence between performance at the daily time step and the model's ability to represent interannual flow variability.

20. Lines 407-408: But at the same time, the MSWEP + ERA5 dataset would still outperform the others if it had been used for calibration and evaluation (if it were available on the same periods), so I am not sure this point holds. It is true that consistency is important, but perhaps the gain would be much more if the resolution was also highly increased.

We thank the reviewer for this thoughtful observation. We agree that MSWEP + ERA5, if available over the full study period, would likely provide better performance in simulating streamflow than the coarser-resolution reanalyses (ERA-20C and NOAA 20CR), due to both higher spatial resolution and more accurate meteorological inputs. However, our point in this paragraph is to highlight a practical and robust conclusion within the constraints of using centennial reanalysis products: namely, that the consistency between the forcings used in calibration and simulation plays a key role in maintaining simulation quality, even when spatial resolution is lower.

We fully acknowledge the potential benefit of using high-resolution downscaled versions of centennial reanalyses. However, this is not straightforward. The development of such downscaled products over the full 20th century remains an open challenge. In particular, ensuring spatiotemporal homogeneity and physical consistency of the downscaled fields is difficult due to the lower density and quality of observational constraints in the early part of the century. This raises important questions about the robustness and reproducibility of trends or low-frequency variability derived from such downscaled data.

For this reason, we chose to retain this discussion point in the Conclusion, as it not only summarizes a key finding of the study (regarding the importance of consistency) but also opens up an important research perspective on the development of reliable, century-long high-resolution forcings. We now clarify this more explicitly in the revised conclusion:

An important result to be highlighted is the necessity of ensuring consistency between the climate forcings used for calibration and those used for simulation: the best performance with the ERA-20C forcing was obtained when using parameters calibrated with the same forcing, despite its relatively coarse spatial resolution. This suggests that consistency in meteorological inputs may play a more critical role than spatial resolution when working with centennial reanalyses. Although higher-resolution datasets such as MSWEP would likely yield better performance if available over the full historical period, producing such datasets over the full 20th century requires downscaling methods that raise challenges in terms of temporal consistency and robustness. While downscaling remains a promising direction for improving local-scale performance, particular care must be taken to ensure the robustness of long-term hydrological trends.

21. General comment: How are NOAA and ERA related? i.e. I imagine they must share a lot of the same historical data for the period prior to 1970-ish. It might be good to give more details on these in the data section.

**(This is the same response as the one provided for point 12 raised by Reviewer 2)**

Thank you for this important comment. We have added a paragraph in the "Data" section (subsection 2.1: Climate forcings) highlighting the similarities and differences between the two reanalysis datasets in terms of assimilated data:

**2.1 Climate forcings**

[...]

While the two reanalyses differ in terms of their underlying models and data assimilation techniques, they also diverge in the types of assimilated observations and prescribed forcings. The NOAA 20CR reanalysis is based solely on surface pressure observations from the International Surface Pressure Databank (ISPD; Cram et al., 2015), assimilated into NOAA's Global Forecast System, with sea surface temperature (SST) and sea ice concentration (SIC) prescribed as boundary conditions. In contrast, the ERA-20C reanalysis assimilates surface pressure data from both ISPD and the International Comprehensive Ocean–Atmosphere Data Set (ICOADS; Woodruff et al., 2011), as well as marine wind observations from ICOADS. Additionally, ERA-20C prescribes not only SST and SIC, but also solar radiation, tropospheric and stratospheric aerosols, ozone, and greenhouse gas concentrations.

The explanation of the differences observed is now presented as a perspective for future work in the "Discussion" section:

5.3 Consistency of multi-decadal hydrological variability:

[...]

Moreover, explaining the differences in trends obtained with the two climate forcings requires further investigation, in order to attribute these discrepancies to differences in assimilated data or, for example, to differences in the underlying climatic models.

22. In the Author contribution section, there is PB and OD, but I imagine OD = LO?

Yes, we corrected this error in the revised version of the article.

**Reviewer 2**

The paper seeks to explore the applicability of two long term reanalysis datasets for reconstructing river flows (low, mean and high) across a large sample of European catchments. This is an important topic for understanding variability and change and contextualising emerging trends and will thus be of interest. I enjoyed reading the paper. While supportive of the paper and ultimately I recommend only minor corrections there are some adjustments to structure and a couple of points of clarity that in my mind would make the paper stronger.

Thank you for these positive comments.

1. The title might be reframed to explicitly include the words 'exploring' or 'evaluating' the utility of these datasets across the flow regime. This is ultimately what the paper does. For a full reconstruction additional uncertainties including hydrological model would need to be included and many of the limitations noted in the discussion integrated into the analysis. However, if the aim is to evaluate the utility of these products then the current study design stands.

Thank you for this suggestion. We changed the title of the article to emphasize the length of the reanalyses used and to incorporate the word "explore" (as suggested by Reviewer 1).

The previous title of the article was: "Using global reanalysis and rainfall-runoff model to study multi-decadal variability in catchment hydrology at the European scale."

We propose the following revised title: "Using century-long reanalysis and a rainfall-runoff model to explore multi-decadal variability in catchment hydrology at the European scale."

2. The introduction and literature review provides a nice summary and collection of useful references.

Thank you for these positive comments.

3. Rather than NOAA and ERA please use the full reanalysis product name throughout for clarity.

We changed NOAA and ERA into NOAA 20CR and ERA-20C all over the article and the figure in the new paper version.

4. Line 81 suggest evaluate rather than document

Thank you for this suggestion: we changed this term in the revised version of the article:

The general objective of this paper is to evaluate the ability of such a modeling methodology to identify trends and/or periodicities of catchment hydrology at the European scale despite the coarse spatial resolution of the global reanalyses and the rainfall—runoff modeling uncertainty.

5. In your aims on line 86, what does efficient mean, use of the word here is a little vague.

Thank you for pointing this out. We agree that the term efficient was vague and have clarified its meaning in the revised manuscript. In this context, we use efficient to refer to the ability of the two global reanalyses to provide climate forcings (precipitation and air temperature) that enable hydrological models to reproduce observed streamflow dynamics across a large sample of catchments. This includes both short-term variability (daily flows) and long-term signals (decadal flow anomalies). As highlighted in the conclusion, the evaluation of efficiency is based on the overall model performance for different flow regimes, and also on the robustness of simulations with respect to the consistency between calibration and simulation forcings.

We have reformulated our first research question, changing it from "How efficient are these two global reanalyses in terms of reconstructing catchment hydrology?" to "How well do these two global reanalyses perform in providing climate forcings that enable hydrological models to reproduce observed streamflow, both at daily and decadal timescales?"

6. Given the scale mismatches can you offer a sentence or two on why downscaling or a combination of downscaling and bias correction was not included?

**(This is the same response as the one provided for point 6 raised by Reviewer 1)**

We thank the reviewer for this important comment. Yes, we conducted tests to extrapolate meteorological forcings before model parameter calibration and rainfall-runoff simulations, particularly for high-altitude catchments. These tests aimed to extrapolate precipitation and air temperature based on the difference between the median elevation of each catchment and the median elevation of the meteorological forcing grid cells in each dataset. The results showed an improvement in calibration performance for small mountainous catchments but also led to a decrease in performance for other catchments. Since the performance improvement was not consistent across all catchments, we did not pursue this option further in the article and used the meteorological forcings as they were, without downscaling. By doing so, we assume having lower performance in terms of hydrological simulations at the daily time step, but we maintain the long-term trends in air temperature and precipitation from the considered reanalyses.

However, it is important to note that the use of the snow accumulation and melt model CemaNeige inherently involves a downscaling of meteorological forcings by distributing precipitation and air temperature over five elevation bands (in our case), which are constructed based on the hypsometric curve of the catchment. Thus, the forcings input into the model are considered representative of those observed at the median elevation of the catchment and are then distributed across the five zones according to the gradients described by Valéry *et al.* (2014, https://doi.org/10.1016/j.jhydrol.2014.04.058).

The development of a downscaling method at the scale of all the European catchments used in our study was beyond the scope of our article but represents a natural perspective for future work.

We have added elements to the paper regarding the extrapolation performed using the CemaNeige module in Section 3.1 (rainfall-runoff model):

The CemaNeige module takes into account the hypsometric curve of each catchment to perform a downscaling of meteorological forcings by distributing daily precipitation and air temperatures across five (in our case) elevation bands. Thus, the forcings input into the model are considered representative of the median elevation of the catchment and are then distributed across the five zones according to the gradients described by Valéry et al. (2014).

7. The section on the four criteria used for catchment selection (line 121) could be shortened with the actual criteria introduced as you list them. I found myself wondering what you mean by relatively long series, adequate area etc. Why the threshold of 100km?

We shortened this paragraph by directly stating what we consider to be a sufficiently long streamflow series, a sufficiently large catchment, etc. The catchment area threshold of 100 km2 is not based on an objective criterion: we stated this in the revised version of the article:

Four criteria were applied to select a sub-sample of European catchments. We retained only catchments that met all of the following conditions:

- At least 10 years of daily streamflow data available during the calibration period (1996– 2010);
- At least 10 years of daily streamflow data available during the evaluation period (1982–1995);
- A catchment area larger than 100 km², a subjective threshold applied to ensure compatibility with the daily time step and the spatial resolution of climate forcings;
- An equivalent water storage capacity from upstream dams below 10 mm, calculated using the GRanD dataset (Lehner et al., 2011) as the ratio between total reservoir storage and catchment area (see Delaigue et al., 2025), to limit the influence of human regulation.

After applying these criteria, 2128 catchments were selected.

8. No need for bullets in differentiating catchment set.

We removed the lists in the revised version of the article.

9. The model calibration process is generally well described, however it might be worth noting how parameter sets were identified – what search was used – the default in GR4J package or another approach.

We used the default optimization algorithm included in the airGR R package. This algorithm was specifically designed for GR models (Michel, 1991; Coron *et al.*, 2017). We added this information in the revised version of the article:

The model was implemented using the R (R Core Team, 2020) package airGR (Coron et al., 2017, 2023), using the default optimization algorithm included in the airGR package. This algorithm was specifically designed for GR models (Coron et al., 2017).

10. A single module structure is used across a very diverse catchment set. Some reflection on why and the strengths/weaknesses of this approach in the context of the aim of the paper would be useful in this section.

We thank the reviewer for this relevant comment. The limitation related to the use of a single rainfall–runoff model across a diverse set of catchments is already explicitly discussed in the Discussion section of the manuscript (subsection 5.1), where we acknowledge the value of a multi-model approach to assess structural uncertainty:

The simulation methodology used in this study has several important limitations to be noted. First, it is based on a single conceptual rainfall–runoff model. Using a multi-model approach would make it possible to quantify the uncertainty associated with the model structure in the simulations performed (Wan et al., 2021; Martel et al., 2023; Thébault et al., 2024).

In line with the objectives of our study — which focused on evaluating the potential of global reanalyses to simulate streamflow at large spatial and temporal scales — we opted for a single, widely used conceptual model (GR4J) that is computationally efficient and has proven robust in a

variety of contexts. This choice allowed us to explore a broad ensemble of catchments over long periods of time, which would be computationally prohibitive with more complex models or multimodel frameworks. The results obtained suggest a predominant influence of meteorological forcings on long-term streamflow variability, indicating that model structural choices may play a lesser role in shaping the trends in flow indicators. However, as we also discuss in the Discussion section, this hypothesis still warrants further testing through sensitivity analyses using different calibration strategies and model structures.

Given that this point is already discussed and justified later in the manuscript, we did not modify the Methods section, in order to avoid overloading it with elements that are better addressed in a dedicated critical discussion.

11. Maybe introduce the Wilcoxon rank test in the methods and why it is used.

This test is now introduced in the method section (3.3.1. Periods of daily streamflow evaluation):

To evaluate whether differences in model performance between the simulations were statistically significant, the Wilcoxon rank-sum test (Wilcoxon, 1945) was applied. This non-parametric test is used to compare two independent distributions and does not require the assumption of normality. It is therefore well suited for assessing differences in performance metrics, such as the KGE values, across a large set of catchments.

12. Fig 7 and elsewhere – have you any suggestions as to why the reanalysis datasets diverge at particular points – eg. Western France prior to 1940 while they show comparable performance for more recent periods. Might there be differences in the sea level pressure data assimilated in each?

**(This is the same response as the one provided for point 21 raised by Reviewer 1)**

Thank you for this important comment. We have added a paragraph in the "Data" section (subsection 2.1: Climate forcings) highlighting the similarities and differences between the two reanalysis datasets in terms of assimilated data:

**2.1 Climate forcings**

[...]

While the two reanalyses differ in terms of their underlying models and data assimilation techniques, they also diverge in the types of assimilated observations and prescribed forcings. The NOAA 20CR reanalysis is based solely on surface pressure observations from the International Surface Pressure Databank (ISPD; Cram et al., 2015), assimilated into NOAA's Global Forecast System, with sea surface temperature (SST) and sea ice concentration (SIC) prescribed as boundary conditions. In contrast, the ERA-20C reanalysis assimilates surface pressure data from both ISPD and the International Comprehensive Ocean–Atmosphere Data Set (ICOADS; Woodruff et al., 2011), as well as marine wind observations from ICOADS. Additionally, ERA-20C prescribes not only SST and SIC, but also solar radiation, tropospheric and stratospheric aerosols, ozone, and greenhouse gas concentrations.

The explanation of the differences observed is now presented as a perspective for future work in the "Discussion" section:

**5.3 Consistency of multi-decadal hydrological variability:**

[...]

Moreover, explaining the differences in trends obtained with the two climate forcings requires further investigation, in order to attribute these discrepancies to differences in assimilated data or, for example, to differences in the underlying climatic models.

13. In the discussion the limitation might be framed better in the context of the aims of the study, i.e. Full exploration of these aspects was not the purpose of the study but rather to evaluate the input datasets.

Thank you for this suggestion. We have clarified the structure of the discussion to better distinguish between the aspects that directly address the study's main objective — the evaluation of global reanalysis datasets as inputs for catchment-scale hydrological modeling — and those that go beyond it. In particular, we now explicitly state that while the detailed quantification of uncertainties related to modeling choices (e.g. model structure, objective function) is outside the scope of the study, these elements are briefly discussed to provide context and help interpret the results (in section 5.1). A dedicated introductory paragraph was added at the beginning of Section 5.1 to frame this part of the discussion accordingly:

The primary aim of this study was to evaluate the suitability of two global reanalyses as inputs for reconstructing catchment-scale hydrology through conceptual rainfall—runoff modeling. To this end, the methodological framework was deliberately kept simple and consistent across catchments, focusing on the effects of the input forcings rather than the modeling choices themselves. Consequently, aspects such as the choice of a single hydrological model, the use of a fixed objective function, or the daily temporal resolution were not explored in depth. Although a full quantification of the uncertainties associated with these methodological decisions lies beyond the scope of this study, it is nevertheless important to acknowledge and briefly discuss these limitations, as they can influence the interpretation of the results.

14. Results are presented throughout the discussion section. It would be better for the reader and the clarity of the paper if these were in the results section.

Thank you for this remark. To improve the clarity and structure of the manuscript, the three figures that were previously included in the discussion section have been either moved to the appendix (e.g. former Figures 11 and 12 are now Figures B and C) or removed (former Figure 10). The corresponding text in the discussion has been revised accordingly to summarize the findings more concisely while referring to the figures in the appendix when necessary.

15. The conclusion is rather like a discussion to me. It would be better if the core research questions were discussed in the discussion section and then more concise conclusions drawn. This would help the sharpness and clarity of the paper.

We thank the reviewer for the suggestion. In the revised manuscript, we removed the detailed restatement of the research questions from the conclusion and shortened the section to make it more concise and focused.

**References**

- Coron, L., Thirel, G., Delaigue, O., Perrin, C. & Andréassian, V. (2017). The suite of lumped GR hydrological models in an R package. Environmental Modelling & Software 94: 166-71. https://doi.org/10.1016/j.envsoft.2017.05.002.
- Dawdy, D. R. & Bergmann, J. M. (1969). Effect of Rainfall Variability on Streamflow Simulation. Water Resources Research 5, no 5): 958-66. https://doi.org/10.1029/WR005i005p00958.
- Martel J-L, Arsenault R, Lachance-Cloutier S, Castaneda-Gonzalez M, Turcotte R, Poulin A. 2023. Improved historical reconstruction of daily flows and annual maxima in gauged and ungauged basins. Journal of Hydrology, 129777. <a href="https://doi.org/10.1016/j.jhydrol.2023.129777">https://doi.org/10.1016/j.jhydrol.2023.129777</a>.
- Michel, C. (1991). Hydrologie Appliquée Aux Petits Bassins Ruraux (Applied Hydrology for Small Catchments). Internal Report (Cemagref Antony, France).
- Oudin, L., Perrin, P., Mathevet,T., Andréassian,V. & Michel, C. (2006). Impact of biased and randomly corrupted inputs on the efficiency and the parameters of watershed models. Journal of Hydrology 320, no 1-2: 62-83. https://doi.org/10.1016/j.jhydrol.2005.07.016.
- Thébault C, Perrin C, Andréassian V, Thirel G, Legrand S, Delaigue O. 2024. Multi-model approach in a variable spatial framework for streamflow simulation. Hydrology and Earth System Sciences. Copernicus GmbH, 28(7): 1539–1566. https://doi.org/10.5194/hess-28-1539-2024.
- Valéry, A., Andréassian, V. & Perrin, C. (2014). 'As simple as possible but not simpler': what is useful in a temperature-based snow-accounting routine? Part 2 Sensitivity analysis of the Cemaneige snow accounting routine on 380 catchments. Journal of Hydrology 517: 1176 87. <a href="https://doi.org/10.1016/j.jhydrol.2014.04.058">https://doi.org/10.1016/j.jhydrol.2014.04.058</a>.
- Wan Y, Chen J, Xu C-Y, Xie P, Qi W, Li D, Zhang S. 2021. Performance dependence of multi-model combination methods on hydrological model calibration strategy and ensemble size. Journal of Hydrology, 603: 127065. https://doi.org/10.1016/j.jhydrol.2021.127065.

---

## Author Response (AR2)

**Reviewer 1**

I think the authors have done a very good job in responding to the comments and that the current version is a solid standalone paper that will be of interest to the community. I only have 2 small minor (perhaps technical) comments that I think can be handled at the editorial board level:

Thank you for this positive feedback on our work.

Line 199: I would remove "Comparison not shown in this paper". I understand the rationale behind it, but the sentence sounds like the original KGE was used to compare to other studies but that the authors are not showing these comparisons. It is best to simply state the reason.

Thank you for this suggestion. The sentence was revised to remove "Comparison not shown in this paper" and now simply states the rationale for using the 2009 version of the KGE:

We used the 2009 version of the KGE in order to allow comparison of model performance and parameters with other similar studies conducted on sub-samples of European catchments.

From the previous round of reviews, there was this comment (page 4 of the response to reviewers document):

"General comment: It would be good to have a series of simulations for which only the ones that obtained KGE in calibration/validation above a certain quality threshold are preserved. Ex KGE > 0.7 for NOAA/NOAA and ERA/ERA (not really useful to do ERA5+MESWEP/NOAA(ERA). This would ensure results are not negatively affected by poorly modelled basins/models, as we have some that have quite low KGE values that contribute to the detailed variability results, and are perhaps not as trustworthy."

**The authors responded this:**

"Following the reviewer's suggestion, a complementary analysis was performed by retaining only the catchments with calibration/validation KGE values above 0.7 for the NOAA/NOAA and ERA/ERA simulations. The figures corresponding to this subset (equivalent to Figures 6 to 8 in the manuscript) show very similar spatial patterns and variability ranges to those already presented. Therefore, to avoid overloading the article, these additional figures were not included in the revised version. However, this sensitivity test confirms that the conclusions drawn are not significantly influenced by poorly modelled catchments."

I think that this should at least be stated somewhere in the text, that the test was done. I fear that other readers might have the same questions and if there is no text to answer it, it will remain a question mark on the papers' methodology.

Thank you for this suggestion. A sentence has been added in Section 5.3 of the Discussion ("Consistency of multi-decadal hydrological variability") to indicate that a complementary analysis was performed by retaining only catchments with calibration and validation KGE values above 0.7. As mentioned in the text, this analysis yielded very similar results to those already presented, confirming that the conclusions are not significantly affected by poorly modelled catchments.

In addition, a complementary analysis was performed by retaining only catchments with calibration and validation KGE values above 0.7. The results were very similar to those presented in Figures 6 to 8, confirming that the conclusions are not significantly affected by poorly modelled catchments.

Otherwise, the paper seems solid to me and ready to be published.

Thank you for this positive feedback on our work.